# Mutant p53 Mediates Sensitivity to Cancer Treatment Agents in Oesophageal Adenocarcinoma Associated with MicroRNA and SLC7A11 Expression

**DOI:** 10.3390/ijms22115547

**Published:** 2021-05-24

**Authors:** Ann-Kathrin Eichelmann, George C. Mayne, Karen Chiam, Steven L. Due, Isabell Bastian, Frederike Butz, Tingting Wang, Pamela J. Sykes, Nicholas J. Clemons, David S. Liu, Michael Z. Michael, Christos S. Karapetis, Richard Hummel, David I. Watson, Damian J. Hussey

**Affiliations:** 1Flinders Health and Medical Research Institute—Cancer Program, Flinders University, Bedford Park, Adelaide, SA 5042, Australia; george.mayne@flinders.edu.au (G.C.M.); Karen.Chiam@nswcc.org.au (K.C.); steven.due@flinders.edu.au (S.L.D.); bell.bastian@hotmail.com (I.B.); frederike.butz@charite.de (F.B.); tingting.wang@flinders.edu.au (T.W.); pam.sykes@flinders.edu.au (P.J.S.); michael.michael@flinders.edu.au (M.Z.M.); c.karapetis@flinders.edu.au (C.S.K.); david.watson@flinders.edu.au (D.I.W.); 2Department of General, Visceral and Transplant Surgery, University Hospital of Münster, Waldeyerstrasse 1, 48149 Münster, Germany; 3Department of Surgery, Flinders Medical Centre, Bedford Park, Adelaide, SA 5042, Australia; 4Peter MacCallum Cancer Centre, 305 Grattan St, Melbourne, VIC 3000, Australia; Nicholas.Clemons@petermac.org (N.J.C.); liu.davidsh@gmail.com (D.S.L.); 5Sir Peter MacCallum Department of Oncology, The University of Melbourne, Parkville, VIC 3010, Australia; 6Department of Surgery, Austin Health, Heidelberg, VIC 3084, Australia; 7Department of Gastroenterology, Flinders Medical Centre, Bedford Park, Adelaide, SA 5042, Australia; 8Department of Medical Oncology, Flinders Medical Centre, Bedford Park, Adelaide, SA 5042, Australia; 9Department of Surgery, University Hospital of Schleswig-Holstein, Ratzeburger Allee 160, 23538 Lübeck, Germany; Richard.Hummel@uksh.de

**Keywords:** radioresistance, chemoresistance, oestrogen receptor modulator, oesophageal cancer, miRNA, SLC7A11, xCT, ribosome, reactive oxidative species, miR-27a-3p

## Abstract

*TP53* gene mutations occur in 70% of oesophageal adenocarcinomas (OACs). Given the central role of p53 in controlling cellular response to therapy we investigated the role of mutant (mut-) p53 and SLC7A11 in a CRISPR-mediated JH-EsoAd1 *TP53* knockout model. Response to 2 Gy irradiation, cisplatin, 5-FU, 4-hydroxytamoxifen, and endoxifen was assessed, followed by a TaqMan OpenArray qPCR screening for differences in miRNA expression. Knockout of mut-p53 resulted in increased chemo- and radioresistance (2 Gy survival fraction: 38% vs. 56%, *p* < 0.0001) and in altered miRNA expression levels. Target mRNA pathways analyses indicated several potential mechanisms of treatment resistance. *SLC7A11* knockdown restored radiosensitivity (2 Gy SF: 46% vs. 73%; *p* = 0.0239), possibly via enhanced sensitivity to oxidative stress. Pathway analysis of the mRNA targets of differentially expressed miRNAs indicated potential involvement in several pathways associated with apoptosis, ribosomes, and p53 signaling pathways. The data suggest that mut-p53 in JH-EsoAd1, despite being classified as non-functional, has some function related to radio- and chemoresistance. The results also highlight the important role of SLC7A11 in cancer metabolism and redox balance and the influence of p53 on these processes. Inhibition of the SLC7A11-glutathione axis may represent a promising approach to overcome resistance associated with mut-p53.

## 1. Introduction

Oesophageal adenocarcinoma (OAC) carries a poor prognosis and is one of the deadliest cancers worldwide because of both its aggressive nature and the lack of early symptoms resulting in delayed diagnosis. Despite the introduction of multimodal therapy including irradiation, chemotherapy, and surgery, 5-year survival rates have remained poor, and range between 15% and 25% [1]. The poor outcomes for OAC are primarily due to patients presenting with the disease at an advanced stage when curative treatments are less effective [2,3]. Patients with locally advanced disease are treated with neoadjuvant chemoradiotherapy (nCRT) before surgery [4]. However, due to intrinsic or acquired therapy resistance, only around 30% of patients have a complete pathological response [5]. Non-responders are at risk of both the side effects from the neoadjuvant treatment, and the effects of delaying surgery [6,7]. Furthermore, the mechanisms by which OACs acquire therapy resistance are poorly understood, and there are to date no reliable biomarkers that can be used for routine clinical use to improve the personalization of treatment. In regard to this, considerable attention has been paid to the tumour-suppressor gene *TP53*, which is the most frequently mutated gene across all cancers [8].

*TP53* gene mutations occur in 70% of OACs and contribute to radio- and chemoresistance, resulting in decreased overall patient survival [9,10,11,12]. As an important checkpoint protein, p53 either functions through transcriptional control of target genes or by influencing multiple response pathways [13]. Upon DNA damage, e.g., generated by ionizing radiation, p53 is stabilized through post-translational modifications resulting in accumulation of intracellular p53. Subsequently, increased p53 levels trigger a variety of downstream signaling pathways that mediate the cellular stress response [14].

*TP53* mutations deprive the tumour-suppressing function of the wild-type (wt) and can also give rise to new gain-of-function oncogenic activities, resulting in increased therapy resistance [15,16]. However, acquisition of chemo- and radioresistance related to *TP53* mutations is complex, and can occur at many other levels within the p53 network. Therapies aimed at either restoring wt-p53 activity or inhibiting mutant (mut-) p53 oncogenic activity therefore offer promising approaches in the treatment of cancer. For example, the mut-p53-reactivating compound APR-246 reactivates p53 signaling, restores anti-tumour activity, and re-sensitizes tumour cells to cisplatin and 5-fluorouracil (5-FU) [17,18]. APR-246 does this by triggering lipid peroxidative cell death via the depletion of glutathione, and this effect is modulated by the cystine-glutamate antiporter SLC7A11 [19]. This antiporter promotes cystine uptake and glutathione biosynthesis, resulting in the protection from oxidative stress caused by ionizing radiation and chemotherapeutic agents. Consistent with this, higher levels of SLC7A11 were associated with chemo- and radioresistance, and sensitivity could be restored via SLC7A11 knockdown [19,20,21,22]. As critical modulator of intracellular redox balance, targeting SLC7A11 is considered a promising therapeutic opportunity for cancer treatment [20].

MiRNAs represent another important component in the complex regulatory network of p53, cancer, and therapy response. More than 20 miRNAs have been found to function as direct negative regulators of p53, while other miRNAs are known to target key regulators of p53. Mut-p53 has also been reported to regulate several miRNAs [23]. The important role of miRNAs in the response to radiotherapy [24] as well as chemotherapy [25,26] for cancer treatment is well established.

In the current study, we investigated the role of mut-p53 on the response to irradiation and various cytotoxic agents in a mut-p53-knockout (KO) cell line model. In addition to the conventional chemotherapy agents, cisplatin and 5-FU, cell survival was also assessed following treatment with the oestrogen receptor (ER) modulators 4-hydroxytamoxifen and endoxifen. These agents were included because we previously observed that OAC cell lines express ERα and ERβ, and that tamoxifen has cytotoxic effects in OAC cell lines [27,28]. Given its key role in modulating therapy responses, *SLC7A11*-knockdown was performed to determine whether radiosensitivity could be restored. The Barrett-associated adenocarcinoma cell line JH-EsoAd1 in this model has a missense mutation in TP53 [29,30], which has been knocked out by CRISPR/Cas9 technology, resulting in three control parental lines retaining endogenous mut-p53 (referred to as “Parentals”) and three mut-p53 KO clones (referred to as “p53-KO”). We also investigated the effect of p53-KO and *SLC7A11*-knockdown on the expression of a panel of miRNAs to provide insight into the possible mechanisms involved.

## 2. Results

### 2.1. Effect of p53 Knockout on Cell Survival after Ionising Irradiation

We confirmed by Western blot that p53 was undetectable in p53-KO compared to parental cells (Figure 1A), and that there was no difference in the ability of mock-irradiated parentals and p53-KO cells to form colonies (*p* = 0.538) (example in Figure 1C, column “non-irradiated”, Appendix A). p53-KO resulted in significantly increased radioresistance: following 2 Gy irradiation, the survival fractions (SF) of the p53-KO and the parental cells were 56% (±SD 6.2%) and 38% (±SD 2.6%) respectively (*p* < 0.0001; Figure 1B).

### 2.2. Effect of p53 Knockout on Cell Survival after Drug Treatment

Dose response curves for the drug treatments are presented in Figure 2. The IC_50_ (50% inhibitory concentrations) of the p53-KO cells derived from these response curves were higher for all four drugs (cisplatin, 5-FU, 4-hydroxytamoxifen, and endoxifen; lower panel, Figure 3) compared to the parental cells. The cell lines were then treated at the IC_50_ of each drug in two further independent experiments. In all experiments there were higher proportions of viable cells and lower proportions of early and late apoptotic p53-KO cells compared with the parental cells for all four drugs. The effects of p53-KO on the increased proportion of viable cells were significant after treatment with cisplatin (14%, *p* = 0.019), 5-FU (18%, *p* = 0.010), and 4-hydroxytamoxifen (20%, *p* = 0.006), but not with endoxifen treatment (11%, *p* = 0.082, Figure 3, Appendix A).

### 2.3. Alterations in MiRNA Expression in JH-EsoAd1 Cells after p53 Knockout

The expression of 111 miRNAs were measured by OpenArray^®^ in parental and p53-KO cells. Eighty-two of the 111 miRNAs were detected in all six cell lines (Appendix A. Of these, 15 miRNAs were differently expressed after KO of p53 at an estimated false discovery rate (FDR) of 10%; five of these were increased, and ten decreased (Table 1, and Appendix A for further details of these miRNAs). Several of these miRNAs have been reported to be regulated by p53, or are regulators of p53. Among these miRNAs, miR-27a-3p was found to be the most significantly increased miRNA by p53-KO, while miR-324-3p was the most significantly decreased miRNA.

### 2.4. Pathway Analyses

We then investigated the potential role of the most significantly increased (miR-27a-3p) and the most significantly decreased (miR-324-3p) miRNAs, as well as the set of *all* miRNAs that were differentially expressed (either increased or decreased) in regulating pathways that might mediate the treatment response.

#### 2.4.1. Pathway Analyses of Increased miRNAs

miRTarBase revealed 431 different validated mRNA targets of miR-27a-3p (Appendix A). Among these, *SLC7A11* and *TP53* were identified as validated targets of miR-27a-3p. All validated targets were then used for pathway analyses using Innate DB. Gene Ontology analysis demonstrated that apoptosis, autophagy, proliferation, and response to DNA damage with respect to ionizing radiation were among the top enriched terms for biological process (Appendix A). Moreover, biological pathway analysis demonstrated that the validated targets were mainly enriched in lipid metabolism and in cellular processes such as cell cycle and gene expression. mRNA targets of miR-27a-3p were also involved in the p53 pathway or were identified as direct p53 effectors (indicated with ***** in Table 2, and Appendix A).

Besides miR-27a-3p, four more miRNAs (miR-24-3p, miR-130b-3p, miR-181a-5p, miR-185-5p) were significantly increased after p53-KO. Validated mRNA targets of these miRNAs are presented in Appendix A. *TP53* was identified as validated target of miR-24-3p and miR-185-5p; *SLC7A11* as target of miR-181a-5p.

The enriched Gene Ontology functions of the validated targets included general cellular processes such as regulation of cell proliferation and growth, cell cycle, gene expression, apoptosis, and autophagy, as well as specific p53 pathways. The latter, for example, included “p53 binding”, intrinsic apoptotic signaling pathway (in response to DNA damage) by p53 class mediator, “positive regulation of DNA damage response, signal transduction by p53 class mediator”, and “DNA damage response, signal transduction by p53 class mediator resulting in cell cycle arrest” (Appendix A). The biological pathway analysis confirmed these observations: the validated targets were significantly enriched in regulation of general cellular processes, lipid metabolism, numerous different pathways such as WNT, mTOR, VEGF, TGF-beta signaling pathways (pathways that are indirectly regulated by p53) as well as in direct p53-dependent pathways (Table 2, Appendix A). These findings indicate that the five miRNAs that were found to be increased after p53-KO are directly involved in the regulation and function of p53, as well as in several pathways that are known to modulate therapy response.

#### 2.4.2. Pathway Analyses of Decreased MiRNAs

miRTarBase revealed 340 different validated mRNA targets of miR-324-3p (Appendix A). Several of these, such as RP *LP0*, *LP1*, *L4*, *L10*, *L10A*, *L31*, *L35A*, *S3**,*
*S5,* and *S8*, are genes that encode for ribosomal proteins. InnateDB ontology and pathway analysis of miR-324-3p revealed that the mRNA targets are predominantly involved in ribosome pathways (Appendix A).

In addition to miR-324-3p, nine other miRNAs were found to be decreased after p53-KO (Table 1). Five (miR-26a-5p, miR-146b-5p, miR-320a-3p, miR-324-5p and miR-328-3p) of these 10 miRNAs are known to target *AGO1* (*EIF2C1*), which encodes a member of the argonaute family of proteins. These proteins play a central role in RNA-mediated gene silencing via mRNA cleavage or translation inhibition [39].

Gene Ontology analysis demonstrated that similar to the enriched terms that were found for targets of increased miRNAs as described in Section 2.4.1, general cellular processes such as regulation of cell proliferation and growth, cell cycle, gene expression, apoptosis, and autophagy as well as specific p53 pathways were among the top enriched terms (Appendix A). The biological pathway analysis indicated as well that the mRNA targets of the ten decreased miRNAs were involved in general cellular processes, specific p53 and lipid metabolism, and ribosome pathways (Table 3).

### 2.5. Rationale for Studying SLC7A11, a Key Regulator of the Antioxidant Response

Given the central role of SLC7A11 in modulating therapy responses via the regulation of intracellular redox balance [40], the targeting of SLC7A11 represents a promising approach to overcome radio- and chemotherapy resistance. Previous work has shown that SLC7A11 expression is increased in p53-KO cells compared to their mut-p53-expressing parental cells, which suggests that the p53-KO cells may have an enhanced ability to protect themselves from reactive oxidative species (ROS) induced by irradiation and antineoplastic agents [19]. Thus, the higher SLC7A11 expression levels in p53-KO cells may have contributed to the increased drug and radiation resistance that we observed. Three miRNAs (miR-26b-5p, -27a-3p -181a-5p) that were differently expressed between parental and p53-KO cells are listed in miRTarBase as direct regulators of *SLC7A11* expression. Further, pathway analysis indicated that two of the differentially expressed miRNAs (miR-27a-3p, -130b-3p) are involved in the regulation of lipid metabolism pathways associated with the oxidative stress response, such as “Regulation of lipid metabolism by Peroxisome proliferator-activated receptor alpha (PPARalpha)”, via direct targeting of *PPARG*. *PPARG* is also regulated by the antioxidant transcription factor NRF2, which is activated by various oxidative stressors [41] and known to control *SLC7A11* expression levels [19,42]. The results suggest a complex and interdependent regulatory network of the cellular antioxidative system, and highlight the potential relevance of the oxidative stress response, with its key element SLC7A11, in our cell line model. Therefore, we investigated whether knocking down *SLC7A11* expression could restore radiosensitivity in p53-KO cells.

### 2.6. SLC7A11 Knockdown Restores Radiosensitivity of p53-KO Cells

It has previously been demonstrated that p53-KO cells have higher expression levels of SLC7A11 than their parental cells [19]. We confirmed this observation in our laboratory (Figure 4A), before proceeding with the *SLC7A11* knockdown experiments using siRNA. Western blot analysis demonstrated that siRNA knockdown of *SLC7A11* in p53-KO cells resulted in a 75% reduction in the level of SLC7A11 expression (Figure 4B).

Following 2 Gy irradiation, the control (non-binding siRNA) treated p53-KO cells had higher radioresistance than the parental control cells (SF 73% vs. 45%, *p* = 0.0088), which is consistent with the radioresistance of un-transfected cells in Figure 1B above. siRNA knockdown of *SLC7A11* decreased the radioresistance in the p53-KO cells compared to the cells treated with the non-binding siRNA (46% vs. 73% SF, *p* = 0.0239), restoring the radiosensitivity of the p53-KO cells to the levels observed in the parental cells (Figure 4C).

### 2.7. Alterations in MiRNA Expression and Predicted Pathways Resulting from SLC7A11 Knockdown

We also profiled miRNA expression in the p53-KO cells after *SLC7A11* knockdown and found four miRNAs that were differentially expressed at an estimated false discovery rate of 10% (Table 4, and Appendix A).

Following *SLC7A11* knockdown in p53-KO cells, miR-331-3p was the most significantly decreased miRNA. miRTarBase revealed 408 validated mRNA targets of miR-331-3p including *RPLP0*, *RPL7A*, *RPL29*, *RPL34*, *RPL36A*, *RPS2*, *RPS3*, *RPS9*, *RPS12*, *RPS14*, *RPS27*, *MRPS26* (Appendix A). Subsequent, Gene Ontology analysis demonstrated that these targets were among the top enriched terms associated multiple times with ribosomal proteins as well as with specific p53-associated terms (e.g., intrinsic apoptotic signaling pathway by p53 class mediator, DNA damage response, signal transduction by p53 class mediator resulting in cell cycle arrest). Pathway analysis also predicted strong involvement in several pathways associated with ribosomes as well as with p53 (e.g., stabilization of p53, p53-dependent G1 DNA damage response, p53-dependent G1/S DNA damage checkpoint; Appendix A).

miR-30a-5p was (by *p*-value) the most significantly increased miRNA by *SLC7A11* knockdown in p53-KO cells. Among 736 validated mRNA targets of miR-30a-5p, *SLC7A11* and *TP53* were identified, indicating a possible feedback loop (Appendix A). Biological pathway analysis indicated that miR-30a-5p targets were potentially involved in cellular processes such as apoptosis, cell cycle, and gene expression as well as in the p53 signaling pathway or identified as direct p53 effectors (Table 5). Furthermore, miR-1274A and miR-1274B were significantly increased following *SLC7A11* knockdown. miRTarBase listed only eight mRNA-validated targets for miR-1274A, and three validated mRNA targets for miR-1274B. Gene Ontology analysis demonstrated that actin cytoskeleton organization, positive regulation of cell adhesion, regulation of actin filament polymerization, and actin binding were among the top enriched terms for biological process associated with the targets of miR-1274A and miR-1274B. Biological pathway analysis identified EPH-Ephrin signaling and regulation of actin cytoskeleton as the top enriched pathways (Appendix A).

### 2.8. Interaction of MiRNAs with Oestrogen Signaling

We identified three miRNAs that were differentially expressed after p53-KO, that were directly associated with pathways that mediate the effects of ER modulators. “Plasma membrane oestrogen receptor signaling” and the gene *ESR1* were identified as modified by miR-130b-3p, and “validated nuclear oestrogen receptor alpha network” containing the *ESR1* and *ESR2* gene was identified as targeted by miR-26a-5p and miR-140-3p. Furthermore, miR-30a-5p—differentially expressed after *SLC7A11* knockdown—was found to be involved in plasma membrane oestrogen receptor signaling via targeting of *ESR2* (Appendix A). Therefore, these miRNAs may have contributed to the effects on cell survival that we observed following treatment with 4-hydroxytamoxifen and endoxifen.

## 3. Discussion

Ionizing irradiation and antineoplastic agents such as cisplatin and 5-FU are key elements in the therapy of OAC. Unfortunately, a large proportion of patients do not benefit from these treatments due to intrinsic or acquired therapy resistance, and the underlying mechanisms are still not well understood. In this regard, considerable attention has been paid to the tumour-suppressor gene *TP53*, the most frequently mutated gene in OAC. *TP53* gene mutations resulting in loss of p53-mediated tumour suppression are associated with radio- and chemoresistance [9,43]. Aside from losing this tumour-suppressive ability of the wt form, mut-p53 proteins can acquire novel oncogenic activity by a gain-of-function mechanism. The majority of cancer-associated mutations in *TP53* are missense mutations, and these are reported to lead to accumulation of mut-p53 protein to high levels, contributing to gain-of-function potential and malignant progression [16,44,45]. Therapies aiming at either reinstating wt-p53 function, or inhibiting mut-p53 oncogenic activity, provide promising approaches in the treatment of cancer.

The current data demonstrate that KO of mut-p53 (p53-KO) results in increased radio- and chemoresistance in OAC cells. Fifteen miRNAs were differentially expressed between parental and p53-KO cells. Validated targets of these miRNAs are enriched in direct p53 pathways, as well as multiple pathways that are controlled by p53, including ribosomal, metabolic, and oxidative phosphorylation pathways, which have been reported to mediate radio- and chemoresistance in cancer [46,47,48,49]. Furthermore, p53-KO cells had higher levels of expression of SLC7A11 than parental cells, and radiosensitivity could be restored via *SLC7A11* knockdown, most likely by rendering p53-KO cells susceptible to oxidative stress. Validated targets of the four miRNAs that were differentially expressed following *SLC7A11* knockdown were enriched in several pathways associated with apoptosis, ribosomes, and p53 signaling pathways.

### 3.1. Effect of p53 Knockout on Cell Survival after Ionizing Irradiation and Drug Treatment

Numerous *TP53* mutations are described for OAC cell lines (summarized in supplementary data of [50]). In our JH-EsoAd1 cell line model, the missense c797G>A mutation in TP53 resulting in an amino acid change of G266E is classified as “non-functional” [29,30]. However, KO of this mut-p53 resulted in increased radioresistance as well as increased resistance to cisplatin, 5-FU, and 4-hydroxytamoxifen, suggesting that cancer cells with p53 missense mutations might retain residual function in relation to cell death activity and respond better to anticancer therapy than p53 null mutations. Studying the impact of *TP53*-KO on response in other oesophageal adenocarcinoma cell lines with *TP53* missense mutations would be a logical step in exploring this further.

Activation of wt-p53 in response to cellular stress, induced for example by radiotherapy or chemical agents, triggers a cascade of numerous cellular effects such as activation of cell cycle arrest, DNA repair, apoptosis, and inhibition of angiogenesis that collectively contribute to tumour growth inhibition and chemo- and radiosensitivity [51]. This cascade can also be regulated by p53-independent pathways. Moreover, p53 also controls “non-canonical” programs, providing more evidence of a complex interplay [13].

### 3.2. Alterations in MiRNA Expression in JH-EsoAd1 Cells after p53 Knockout Followed by Pathway Analysis

miRNAs can regulate p53 expression, either by directly targeting p53 or by targeting key regulators of p53, and this can contribute to therapy resistance [24,52,53,54,55]. On the other hand, miRNA expression levels can be regulated by mut-p53 and this is also associated with oncogenic functions [23]. In our study, we found 15 miRNAs differentially expressed between parental and p53-KO cells. Among these, mir-27a-3p was the most increased miRNA following p53-KO, and miR-324-3p the most decreased miRNA.

#### 3.2.1. Pathway Analyses of Increased MiRNAs

mir-27a-3p has been reported to be an oncoMir which is overexpressed in several cancer types such as oesophageal [56,57], gastric [58,59], ovarian [60], and breast cancer [31,61,62]. mir-27a-3p has also been reported to act as a tumour suppressor in other cancers [63,64,65]. This implies that miR-27a-3p may function as an oncogene or a tumour suppressor depending on the type of cancer. The important role of mir-27a-3p in tumour biology, cancer cell proliferation, apoptosis, and drug metabolism and resistance is well recognized [66,67]. For example, it has been reported that down-regulation of miR-27a re-sensitized esophageal squamous cell carcinoma [56] and breast cancer cells [62] to antineoplastic agents.

Several of the pathways that we identified as enriched in validated targets of miR-27a-3p are known to play an important role in mediating responses to treatment. For example, we identified a variety of different pathways all associated with lipid metabolism, which were also identified in our previous study of intrinsic radiation and drug resistance in a panel of eight OAC cell lines [50]. Other authors have associated lipid metabolic pathways with cancer cell survival and radiotherapy response, and suggested that these pathways are potentially promising targets for anti-cancer strategies [68,69]. Validated targets of mir-27a-3p such as *APC* were significantly enriched in the WNT signaling pathway (by both Gene Ontology and biological pathway analysis), which exerts crucial roles in tumorigenesis and therapy response. For example, Zhang et al. observed that increased expression levels of miR-27 activated the WNT pathway via *APC* in gastric cancer cells, resulting in the promotion of epithelial-mesenchymal transition and metastasis [58].

Furthermore, several other targets of mir-27a-3p (MET, EGFR, FOXO1, and KRAS) are reported to have an impact on tumorigenic characteristics of different cancers such as non-small cell lung cancer (MET and EGFR [70]), renal cell carcinoma (EGFR [64]), hepatocellular carcinoma (FOXO1 [71]), and oesophageal carcinoma (KRAS [57]). However, perhaps the most striking finding is that *TP53* itself was identified as a target of miR-27a-3p. Towers et al. demonstrated that miR-27a-3p binds to the p53 3′-UTR in breast cancer cells [31]. Pathways analysis revealed p53-associated pathways such as “p53 pathway” and “direct p53 effectors”. It is therefore possible that the increased expression levels of miR-27a-3p following p53-KO resulted in a direct influence on p53 signaling pathways, suggesting a possible p53 feedback loop in our cell line model. Both positive and negative feedback loops involving direct p53 effectors have been described [72]. Additionally, Wang et al. demonstrated that mut-p53 binds to the miR-27a promoter and transcriptionally inhibits its expression [73]. This mechanism may explain the decreased miR-27a-3p levels in the parental compared to the p53-KO cells.

Gene Ontology and biological pathway analysis of all five miRNAs that were increased in p53-KO cells indicated the potential involvement of target mRNAs in specific p53 pathways as well as in pathways that are indirectly controlled by p53 such as WNT, mTOR, VEGF, TGF-beta, and apoptotic signaling pathways. These pathways have also been reported to impact on radio- and chemoresistance [43,74,75,76], suggesting that these miRNAs may be important for the irradiation and drug effects we observed in our model. Moreover, various pathways that were significantly enriched in this cell line model such as lipid metabolism, apoptosis, PTEN-dependent cell cycle arrest and apoptosis, plasma membrane oestrogen receptor signaling, and p53 signaling pathway were also identified in our previous analysis of intrinsic radiation and drug resistance in a panel of eight OAC cell lines [50].

#### 3.2.2. Pathway Analyses of Decreased MiRNAs

miR-324-3p is involved in several types of cancer such as nasopharyngeal carcinoma (NPC) [77], lung [78,79], and colon cancer [80], and it is also known to impact on radioresistance [74]. In our study, miR-324-3p was the most significantly decreased miRNA in the resistant p53-KO cells. In line with these findings, Li et al. observed a significant down-regulation of miR-324-3p in an established radioresistant NPC cell line compared to its parental cells [74]. Xu et al. reported significantly reduced expression of miR-324-3p in tissue of NPC patients suffering from radioresistance compared to radiosensitive patients [77]. Our observation and the findings from these studies suggest that down-regulation of miR-324-3p is associated with therapy resistance in several types of cancer, and that miR-324-3p therefore has potential as a predictive marker for therapy response in OAC.

Gene Ontology and pathway predictions indicated that the vast majority of mRNA targets of miR-324-3p are involved in ribosomal pathways including large numbers of ribosomal protein (RP)-encoding genes. miRNAs have reported roles in the regulation of translation and expression of RPs. The in-silico analysis of Reza et al. demonstrated that miR-324-3p along with more than a thousand other miRNAs are involved in the complex regulation of RPs [81]. Besides their primary role in ribosomal assembly and protein translation, RPs have extra-ribosomal functions [82]. For example, *RPS3*, *RPS5*, *RPL31,* and *RPP1* (all targets of the decreased miRNAs in our study) were previously demonstrated to regulate apoptosis (*RPS3*), cell cycle (*RPL31*, *RPS3*, *RPS5*), proliferation (*RPL31*), cell migration and invasion (*RPS3*), and neoplastic transformation (*RPP1*) [82]. In our previous work, we also observed that the targets of miRNAs associated with radiation and cisplatin response were significantly enriched in ribosome pathways [50].

The WNT signaling pathway with its target *WNT2B*, controlled by miRNA-324-3p, could be another contributor to the observed increased radio- and chemoresistance in the p53-KO cells. Li et al. identified that radioresistance of NPC was regulated by the WNT2B signaling pathway, and that WNT2B expression was regulated by miRNA-324-3p binding to the 5′-UTR of *WNT2B* [74].

In addition to miR-324-3p, nine other miRNAs were found to be decreased after p53-KO. Interestingly, five (miR-26a-5p, -146b-5p, -320a-3p, -324-5p, and -328-3p) of these ten miRNAs are known to target *AGO1* (*EIF2C1*), which is associated to the Gene Ontology term “negative regulation of translation involved in gene silencing by miRNA”. AGO1, a member of the argonaute family, represents a core component of the RNA-induced silencing complex, and together with the ribonuclease Dicer is a key component in the processing of mature miRNAs. Via binding to Dicer, AGO1 supports the Dicer-mediated cleavage of pre-miRNAs into mature miRNAs. It is well accepted that p53 governs expression of Dicer, most probably at transcriptional and post-transcriptional levels, whereas de-regulated Dicer levels, depending on cell context, either promote or inhibit tumorigenesis [83]. In the context of our cell line model, absence of p53 and subsequent de-regulated Dicer and miRNA levels may have contributed to the increased drug and radiation resistance.

Furthermore, specific p53 pathways and pathways associated with lipid metabolism and ribosomes (pathways known to effect therapy sensitivity) were identified in the pathway analysis of all ten down-regulated miRNAs, highlighting the key role of these miRNAs in mediating treatment response in OAC.

### 3.3. SLC7A11 Knockdown Restores Radiosensitivity of p53-KO JH-EsoAd1 Cells

As previously demonstrated by Liu et al. [19], our results confirmed that p53-KO cells have higher expression of SLC7A11 than parental cells. *SLC7A11* knockdown via siRNA restored the radiosensitivity of the p53-KO cells, but did not alter sensitivity in the parental cells. The higher expression of SLC7A11 in p53-KO cells, which potentially enhanced the protection from irradiation-induced oxidative stress, could have contributed to the increased radioresistance of p53-KO vs. parental cells. *SLC7A11* knockdown rendered the p53-KO cells susceptible to oxidative stress, resulting in the restoration of radiosensitivity. This is consistent with the reported association between high expression of SLC7A11 and radio- and chemotherapy resistance, and the observation that sensitivity can be restored by *SLC7A11* knockdown [19,20,21,22].

### 3.4. Alterations in MiRNA Expression after SLC7A11 Knockdown Followed by Pathway Analysis

Following *SLC7A11* knockdown in p53-KO cells, miR-331-3p was the most significantly decreased miRNA. Gene Ontology and pathways analyses predicted an involvement in several pathways associated with ribosomes as well as with p53.

miR-30a-5p was the most significantly increased miRNA. *SLC7A11* and *TP53* were both identified as targets of miR-30a-5p, indicating a possible feedback loop. miR-30a is described as a tumour suppressor that governs multiple biological processes, including proliferation, invasion, metastasis, and apoptosis, suggesting an important role in cancer development [84]. This is in line with the findings from our pathway analysis, which revealed a strong association of miR-30a targets with the apoptotic pathways. Many miR-30a-validated targets related to proliferation, apoptosis, and metastasis such as *EYA2, CBX3, IGF1R*, *NOTCH1*, *PIK3CD,* and *RPA1* [84] were also identified in our analysis.

Beyond its function as tumour suppressor, miR-30a is associated with several pathways that regulate the response to antineoplastic agents, which suggests that miR-30a has potential as a therapeutic target in the treatment of cancer. Saad et al. demonstrated that down-regulation of miR-30a-5p improved sensitivity to cisplatin. The authors showed that miR-30a has binding sites in the 3′UTRs of the tumour suppressor genes *BNIP3L*, *PRDM9*, and *SEPT7*, and that these target genes were upregulated following knockdown of miR-30a-5p [85]. *BNIP3L*, associated in our analysis with the pathway “direct p53 effector”, is a pro-apoptotic gene and known as a key regulator of the p53-dependent cell death. Therefore, re-stored radiosensitivity following *SLC7A11* knockdown could have resulted from increased miR-30a levels, leading to initiation of apoptosis.

Besides miR-30a-5p, miR-1274A and miR-1274B were significantly increased following *SLC7A11* knockdown. miRTarBase revealed only a few validated targets of miR-1274A and miR-1274B that were associated with Gene Ontology terms “actin cytoskeleton organization, positive regulation of cell adhesion, regulation of actin filament polymerization and actin binding” as well as “EPH-Ephrin signaling” and “regulation of actin cytoskeleton” as top enriched pathways. Interestingly, SLC7A11 (X_c_^-^) has been reported to interact under glucose-limited conditions with the EPH-Ephrin signaling pathway via increasing ROS levels and subsequent phosphorylation of EPHA2 [86]. It should be noted that miR-1274A and miR-1274B were determined to be fragments of tRNAs, and that their functional role as “ miRNAs” has consequently been questioned [87]. However, there is some evidence that these fragments possibly act similarly to mature miRNAs. Studies have demonstrated in vitro effects associated with promotion of tumorigenesis of cancer cells transfected with miR-1274A and miR-1274B [88,89], leading to the hypothesis that small RNA fragments, occurring through cleavage of mature tRNAs, might gain miRNA-like silencing activity [90,91]. Further, other authors reported SLC7A11-dependent redox-sensitive proteins to be involved in very similar pathways to those that we identified in the pathway analyses of miR-1274A and miR-1274B targets [92]. Together, the observations are consistent with the possible role for miR-1274A and miR-1274B-modulated pathways in our cell line model.

### 3.5. Possible Roles for Oestrogen Receptor Signaling

We identified several miRNAs that were directly associated with pathways mediating the effects of ER modulator treatment, by targeting the genes that encode ERα and ERβ. This most likely contributed to the observed effects on cell survival following treatment with these agents. OAC tissues [93,94] as well as OAC cell lines have been found to express ERα and ERβ [28] and OAC cell lines respond to treatment with ER antagonists [27,95]. ER signaling also contributes to radioresistance [96]. Furthermore, ERα and ERβ, have also been reported to interact with multiple signaling pathways such as MAPK, WNT/β-catenin, EGFR, and p53 [97]. Liu et al. for example demonstrated that ERα directly binds to p53 target gene promoters, resulting in the functional inactivation of p53 and promotion of tumour growth [98]. ER modulators have been described as functioning independently of the ER-pathway by affecting mitochondrial activity and induction of ROS, followed by an antioxidant response. In this regard, a causal link between ROS levels and antihormone resistance, possibly via accumulation of NRF2, has been suggested recently [99,100].

### 3.6. Limitations

Our study of the effect of p53-KO on susceptibility to irradiation was limited to the clonogenic assay, in line with our primary interest on the long-term effect. We acknowledge that inclusion of proliferation assays could have provided additional information about the short-term effects on susceptibility to irradiation.

We were limited to 111 human miRNAs in the OpenArray configuration for this study. While a broader screen might have identified additional miRNAs of interest, we were able to readily identify the important p53-associated functions of the miRNAs included in our study. Our pathway analysis involving experimentally validated miRNA targets indicated a complex and cumulative network-based effect of the differentially expressed miRNAs. The interpretation of our results is potentially limited because we did not perform the in vitro modulation studies of the differentially expressed miRNAs that we identified. A meaningful in vitro study of these differentially expressed miRNAs could be conducted in the future, and would require the expression of several miRNAs to be simultaneously modulated (both up and down-regulated), and the magnitude of modulation should be restrained so as to mimic the fold changes that we observed.

## 4. Methods

### 4.1. Cell Lines and Culture Conditions

We used the mut-p53 JH-EsoAd1 cell line for these experiments, a cell line derived from a patient with Barrett-associated adenocarcinoma, which has a missense c797G>A mutation in TP53 resulting in an amino acid change of G266E [30]. The IARC (International Agency for Research on Cancer) transactivation classification for this mutation is “non-functional” [29]. Three CRISPR-mediated TP53-KO clones (p53-KO), and three control parental lines retaining endogenous mut-p53 (parental polyclonal, parental clonal, and Cas9 only) were studied. TP53-KO was performed by CRISPR/Cas9 technology at the Peter MacCallum Cancer Centre in Melbourne as described previously [17]. Following TP53-KO, the three clones (Clone 1, 7, and 14) with unique InDel mutations did not contain any residual p53 protein. Functional characterization studies (cell viability and cell cycle assays, clonogenic formation) showed that these clones were phenotypically similar to each other and to parental cells [17].

The cells were cultured in (phenol red free) RPMI 1640 medium (Thermo Fisher Scientific, cat #11835055, Scoresby, Australia) supplemented with 10% fetal bovine serum (FBS, Thermo Fisher Scientific, cat #10099141, Scoresby, Australia), 50 U/mL penicillin (Thermo Fisher Scientific, cat #15070063, Scoresby, Australia), 50 µg/mL streptomycin (Thermo Fisher Scientific, cat #15070063, Scoresby, Australia), and 100 µg/mL normocin (InvivoGen, cat #nta-nr-2, San Diego, CA, USA). Subculturing was performed at a confluence of 70%. To facilitate cell detachment and minimize the incubation time with EDTA-trypsin, two extra phosphate buffered saline (PBS) washes including a PBS incubation of 10 min at 37 °C were performed. Cells were cultured in a humidified atmosphere containing 5% CO_2_ at 37 °C.

### 4.2. Irradiation and Clonogenic Survival Assay

Cells were plated in three technical replicates into 6-well plates at 350,000 cells/well and then incubated for 24 h to allow settling. At a confluency of 70–80%, the cells were then mock-irradiated (0 Gy) or irradiated with 2 Gy in a X-Rad 320 irradiator (Precision X-Ray, North Branford, CT, USA). See [50] for further technical details of the irradiation process. After radiation exposure, survival was defined as the ability of the cells to maintain clonogenic capacity, which is considered as gold standard method for in vitro assessment of radiosensitivity [101,102]. Immediately after irradiation, irradiated and mock-irradiated cells were harvested and plated into 6-well plates in triplicates at 500 (0 Gy) or 750 (2 Gy) cells/well (optimal cell seeding numbers and incubation time were defined by performing preliminary studies with a range of cell seeding densities). The culture medium was changed every 3–4 days. After an incubation time of 7 days, colonies were washed with PBS, fixed for 20 min with 10% neutral buffered formalin, stained for 1 h with 0.01% crystal violet (Sigma-Aldrich, # V5265, Castle Hill, Australia), and washed again. The air-dried colonies were then counted using a dissecting microscope. A colony was defined to consist of at least 50 cells. The plating efficiency (PE) was defined as the number of colonies counted/number of cells plated × 100%. The SF was determined by dividing the PE of the irradiated cells by the PE of the control unirradiated cells and then multiplying by 100%. At least four independent experiments with three technical replicates were performed.

### 4.3. Drug Treatment and Apoptosis Assay

To investigate the role of mut-p53 in response to the various treatment agents, parental and p53-KO cells were treated with cisplatin (Hospira, cat #1885A, Melbourne, Australia), 5-FU (Hospira, cat #2587A-AU, Melbourne, Australia), and the oestrogen receptor modulators 4-hydroxytamoxifen (Sigma-Aldrich, cat #T176, Castle Hill, Australia), and endoxifen (Sigma-Aldrich, cat #E8284, Castle Hill, Australia) followed by a cell death analysis via flow cytometry. For all drug-treatment experiments, cells were cultured in RPMI supplemented with 10% charcoal-stripped FBS (Thermo Fisher Scientific, cat #12676029, Scoresby, Australia) and antimicrobials as above, instead of complete FBS to avoid the confounding effects of steroid hormones [103]. In contrast to the irradiation experiments, these investigations were performed in a pooled setting, meaning that the three parental cell lines respectively, and the three p53-KO clones were cultivated separately but pooled for drug treatment and analysis. This approach was chosen after the performance of preliminary experiments performed in a non-pooled setting which demonstrated similar MTS dose response curves for the individual parental cell lines (Appendix A). Briefly, the three parental cell lines or the three p53-KO clones were pooled and plated in three technical replicates into a 12-well plate at 45,000 cells/well (meaning 15,000 cells of parental polyclonal; 15,000 cells of Parental clonal and 15,000 cells of Cas9 only vs. 15,000 cells of Clone 1; 15,000 cells of Clone 7; 15,000 cells of Clone 14). Cells were allowed to attach for 24 h (resulting in a confluency at time of treatment of 20%), medium was then replaced with 100 µL RPMI + 10% charcoal-stripped FBS-containing vehicle or treatment agents which were prepared as described previously [27]. For generating flow-cytometry-based dose-response curves, pooled parental and pooled p53-KO cells were treated in a first step with the different doses of the four drugs (cisplatin/5-FU: 1 µM, 5 µM, 10 µM, 20 µM, 50 µM, and 100 µM; 4-hydroxytamoxifen/endoxifen: 0.1 µM, 1 µM, 5 µM, 10 µM, 20 µM, and 50 µM). After a drug incubation time of 72 h, all cells were harvested (preceded by two extra washes of PBS that were necessary to facilitate cell detachment), spun down, and resuspended in 100 µL binding buffer. Cells were then incubated in the dark with Annexin-V-APC antibody (BD bioscience, cat #550475, North Ryde, Australia). Total of 150 µL binding buffer and 5 µL propidium iodide (PI, Abcam, cat #ab14083, Melbourne, Australia) were added immediately before analysis. Viable (unstained), early apoptosis (Annexin positive, PI negative), and late apoptosis (both Annexin and PI positive) were counted using an Accuri C6 flow cytometer (BD biosciences, North Ryde, Australia). About 2000 events were collected per replicate. Calculation of IC_50_ was determined by fitting a four-parameter logistic curve to the dose-response data from the apoptosis assays (GraphPad Prism, version 6, La Jolla, CA, USA). Viable cells were normalized to their corresponding vehicle control; early and late apoptotic cells of the vehicle control were subtracted from early and late apoptotic-treated cells.

Following calculation of IC_50_, cells were treated at either parentals’ or p53-KOs’ IC_50_ concentrations (cisplatin: 8 µM, 5-FU: 19 µM, 4-hydroxytamoxifen: 7.8 µM, endoxifen: 6.7 µM) in two further independent experiments. For the dose response curves the flow cytometry values were interpolated at the IC_50._ Two further experiments were then performed at the IC_50_. Percent changes in viable, early apoptotic, and late apoptotic cells were averaged across the three experiments and subjected to Student’s *t*-tests.

### 4.4. SLC7A11 Knockdown Using siRNA

For transfection experiments, the media was changed 24 h prior to transfection to antibiotic-free charcoal-stripped FBS. Cells were then transfected using Lipofectamine 2000 (Invitrogen, cat #11668019, Scoresby, Australia) in a pooled setting as described above. After harvesting of the cells, 2.04 × 10^6^ cells were placed in T25 flasks which resulted in 30–50% confluency after 6 h. Cells were transiently transfected with *SLC7A11* siRNA (ON-TARGETplus Human SLC7A11 (23657) siRNA-SMARTpool, Dharmacon/Horizon, cat #L-007612-01-0010, Cambridge, UK) or control siRNA (ON-TARGETplus Non-targeting Pool, Dharmacon/Horizon, cat #D-001810-10-05, Cambridge, UK) to inhibit expression of *SLC7A11* or not, respectively. The final siRNA concentration was 100 nM. Additionally, an untransfected control group with medium and Lipofectamine only was included. The transfection medium was replaced 6 h after transfection. Two independent experiments were performed.

### 4.5. Irradiation and Clonogenic Survival Assay Following SLC7A11 Knockdown

About 24 h following transfection, pooled parental and p53-KO cells were harvested and plated for irradiation as described above. Irradiation and clonogenic assay were then performed as described above.

### 4.6. Statistical Analyses

For statistical analysis of differences between treatment responses, Student’s *t*-test was performed using the GraphPad Prism software (GraphPad Prism, version 6, La Jolla, CA, USA). All data are given as mean ± standard deviation as not stated otherwise. Two-tailed *p* values < 0.05 were considered statistically significant.

### 4.7. Western Blot

Transfected cells (at treatment relevant density) were collected 24 h after transfection, lysed in RIPA buffer (2 mM EDTA, 1% sodium deoxychlorate, 0.1% sodium dodecyl sulfate (SDS), 1% Triton-X, PBS) mixed with protease and phosphatase inhibitor tablets, sonicated on ice, and centrifuged at 10,000× *g* at 4 °C for 20 min. Samples were aliquoted and stored at −80 °C. The protein concentration of each sample was quantified using the EZQ protein quantitation assay (Thermo Fisher Scientific, cat #R33200, Scoresby, Australia). For analysis, 15 µg of protein was loaded onto a 4–20% precast polyacrylamide gel (4–20% Criterion TGX Stain-Free Protein Gel, BioRad, cat #5678094, Gladsville, Australia) and transferred to polyvinyl difluoride membranes with primary antibodies of anti-xCT/SLC7A11 (D2M7A) (Cell Signaling Technology, cat #12691, Danvers, MA, USA) and anti-p53 (Santa Cruz, cat #SC-126, Dallas, TX, USA) in a dilution of 1:1000 followed by a horseradish peroxidase-conjugated secondary antibody (Jackson Immunoresearch Laboratories, Anti-rabbit-HRP: cat #715-035-152, Anti-mouse-HRP: cat #715-035-150, West Grove, PA, USA) in a dilution of 1:3000. Chemiluminescent detection was performed with the ChemiDoc™ MP imaging system (BioRad, Gladsville, Australia) using Clarity Western ECL Substrate (BioRad, cat# 170-5061, Gladsville, Australia). Digital ECL imaging was then performed in an ImageQuant™ LAS 4000 Imager (GE Healthcare, Parramatta, Australia), using the Image Lab™ software (version 4.0, BioRad, Gladsville, Australia) for analysis. For protein quantification, protein expression was normalized to total protein load—removing variations associated with comparing abundance to a single protein—using a previously described method [104]. Briefly, in this method the abundance of the protein of interest was normalized to the total amount of protein in each lane. For this purpose, PVDF membranes were imaged after protein transfer (using a proprietary fluorescent compound), identical areas through the center of each lane from top to bottom were marked and the adjusted volume intensity was measured. A normalization factor was calculated by dividing the intensity of each lane with the intensity of the lane with the highest intensity value on the membrane. After blotting, the total intensity of each band on the blot was then divided by this normalization factor to provide a quantitative estimate [104].

### 4.8. RNA Extraction

Cells were lysed using QIAzol reagent (Qiagen, cat #79306, Chadstone, Australia) and stored at −80 °C until RNA extraction was performed, using the miRNAeasy Kit (Qiagen, #217004, Chadstone, Australia) according to the instructions of the manufacturer. The final RNA solution was stored at −20 °C. The concentration of RNA was quantified by UV spectrophotometry (NanoDrop^TM^ 2000 Spectrophotometer, Thermo Fisher Scientific, Wilmington, DE, USA).

### 4.9. TaqMan^®^ OpenArray^®^ MiRNA Profiling

MiRNA expression analysis was performed using the high throughput TaqMan^®^ OpenArray^®^ (Scoresby, Australia) system. In this study, custom made Primer Pools (RT, Life Technologies, cat #A25630 and PreAmp Primer Pools, #4485255, Scoresby, Australia) and a custom made OpenArray^®^ plate containing assays for 111 human unique miRNAs were used [50]. OpenArray miRNA profiling was performed with untreated parental polyclonal, parental clonal, Cas9 only, Clone 1, Clone 7, and Clone 14 harvested immediately prior to irradiation or drug treatment as well as with parentals vs. p53-KOs in a pooled setting harvested after SLC7A11-knockdown. In this regard, cells were plated under equivalent conditions as described in 4.2–4.5. For each sample, 3.35 μL of RNA, extracted as described above, was reverse transcribed using the TaqMan^®^ microRNA Reverse Transcription Kit (Life technologies, #4366596, Scoresby, Australia) in combination with the Megaplex RT custom primers. RT was followed by cDNA preamplification, carried out by TaqMan PreAmp Reaction Mix (Life technologies, #4488593, Scoresby, Australia) including the custom made 2.5 μL Megaplex PreAmp Primers and 2.5 μL of cDNA. Following preamplification, 4 μL of each sample was diluted at the recommended 1:40 dilution by adding 156 μL RNase-free ultrapure water before mixing with TaqMan OpenArray Real-Time PCR Master Mix (Life Technologies cat #4462164, Scoresby, Australia). The samples were transferred to the 384-well TaqMan OpenArray plate QuantStudio™ 12K Flex OpenArray™ PCR Plates Custom (Thermo Fisher Scientific, cat #4470187, Scoresby, Australia) and PCR runs were performed using the QuantStudio^TM^ 12K Flex Real-Time PCR System at Flinders Genomics Facility, Flinders University.

### 4.10. House Keeping Gene (HKG) Selection

For normalization of the miRNAs, we selected 48 miRNAs as HKGs from the parental and p53-KO cell lines (Appendix A), and 42 miRNAs as HKGs from the SLC7A11-knockdown experiments (Appendix A). There was 93% overlap in the selected HKGs from the SLC7A11-knockdown experiments vs. the parental and p53-KO cell lines. The selection of HKGs was performed by applying a modified version of the method of Bianchi et al. [105], using the following criteria: (i) They were expressed in all samples and at high levels (median Ct < 30); (ii) they were not statistically different in tissue comparisons (Mann Whitney U test, *p* > 0.1); (iii) they were not highly variable (coefficient of variation < 2× standard deviation) and did not contain outliers (samples with levels not within 5-fold of the mean); and (iv) they were correlated at r > 0.7 with the geometric mean of the HKGs.

### 4.11. Sample Comparisons, for Pooling of Cell Lines

Correlation scatter plots were generated between (i) the cell line clones, and (ii) between experiments for the p53-KO analyses, and between experiments for the SLC7A11-knockdown analyses, to determine whether the normalized expression levels of the miRNAs were stable across the cell lines and experiments (Appendix A). We also estimated the overall precision across the cell lines and experiments: (i) For the p53-KO cells, the median variance of the miRNAs (from the cell lines and experiments, and for which 100% of samples amplified) from the mean of the pooled miRNA expression in the p53-KO cells across different experiments was 20.5% (interquartile range (IQR) 10.2–33.8%); (ii) for the parental cells, the median variance of the miRNAs (from the cell lines and experiments, and for which 100% of samples amplified) from the mean of the pooled miRNA expression in the p53-KO cells across different experiments was 22.7% (IQR 11.4–39.6%); (iii) for the p53-KO cells which were treated with a non-binding siRNA, the median variance of the miRNAs (from the two experiments) from the mean of the pooled miRNA expression in the p53-KO cells across different experiments was 8.5% (IQR 3.5–15.3%); (iv) for the p53-KO cells which were treated with *SLC7A11*-knockdown siRNA, the median variance of the miRNAs (from the two experiments) from the mean of the pooled miRNA expression in the p53-KO cells across different experiments was 8.6% (IQR 2.8–15.4%).

### 4.12. Differential Expression Analyses

Differences in the expression of miRNAs were tested using an Empirical Bayes *t*-test (limma package v3.38.3, Bioconductor) [106] using R statistical software v3.5.2, Vienna, Austria) followed by FDR estimation (performed using the method of Storey (2002) [107] in Microsoft Excel (v16.16.24, Redmond, WA, USA)) in pooled parentals vs. p53-KO cells prior to treatment, as well as in p53-KO cells treated with a non-binding siRNA vs. the SLC7A11-knockdown siRNA.

### 4.13. Gene Ontology and Biological Pathway Enrichment Analysis

The database miRTarBase was used for the identification of validated targets of the differentially expressed miRNAs [108,109] (http://mirtarbase.cuhk.edu.cn/php/index.php) (accessed on 18 August 2020) followed by a Gene Ontology and biological pathways analysis via InnateDB (https://www.innatedb.com/) (accessed on 17 December 2020) for the detection of pathways that include a statistically significant number of targets [110]. Gene Ontology and biological pathway terms with a corrected *p*  value ≤  0.05 were considered significantly enriched.

## 5. Conclusions

Our data demonstrate that KO of p53 resulted in increased radio- and chemoresistance, suggesting that patients with a p53 missense mutation are potentially more likely to benefit from chemo- or radiotherapy than patients with a p53 null mutation. Furthermore, KO of p53 resulted in altered expression of miRNAs. Pathway analysis indicated involvement of their mRNA targets in direct p53 pathways and multiple pathways that are controlled by p53 including ribosomal, metabolic, and oxidative phosphorylation pathways which have been reported to mediate radio- and chemoresistance in cancer. *SLC7A11* knockdown restored radiosensitivity in p53-KO cells, possibly via enhanced sensitivity to oxidative stress, and significantly affected miRNA expression. Pathway analysis of the predicted mRNA targets demonstrated involvement in several pathways associated with apoptosis, ribosomes, and p53 signaling pathways. The current data suggests that mut-p53 in JH-EsoAd1, despite being classified as non-functional, has some function related to radio- and chemoresistance. The results also provide promising starting points for a better understanding of the important role of SLC7A11 in cancer metabolism and redox balance and the influence of p53 on these processes. Inhibiting the SLC7A11-gluthatione axis may represent a promising approach to overcoming treatment resistance associated with p53 mutation. Future studies should investigate the functional impact of the identified miRNAs upon *SLC7A11* expression and response to anti-neoplastic agents in this cell line model.

## Figures and Tables

**Figure 1 ijms-22-05547-f001:**
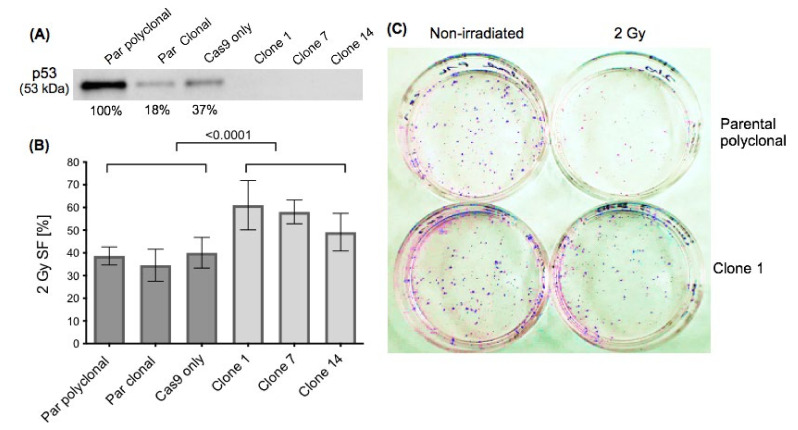
Response to irradiation. (**A**): Western blot demonstrating absence of p53 in the clones. The % value represent the normalized expression (for protein quantification, protein expression was normalized to total protein load, see Section 4.7. for further details). (**B**): 2 Gy survival fractions of the three parental (Par) cell lines (parental polyclonal, parental clonal, Cas9 only) and the three p53-KO clones. The graph shows results from the clonogenic assay from five independent irradiation experiments. (**C**): Example of stained colonies after an incubation period of 7 days. Left column: non-irradiated cells, right column: cells treated with 2 Gy ionizing irradiation. Upper row: parental polyclonal, lower row: Clone 1. Absolute numbers of colonies of the mock-irradiated cells were similar (parental polyclonal: 95 colonies, Clone 1: 102 colonies;). After 2 Gy irradiation, p53-KO Clone 1 formed more colonies (88) than parental polyclonal (53).

**Figure 2 ijms-22-05547-f002:**
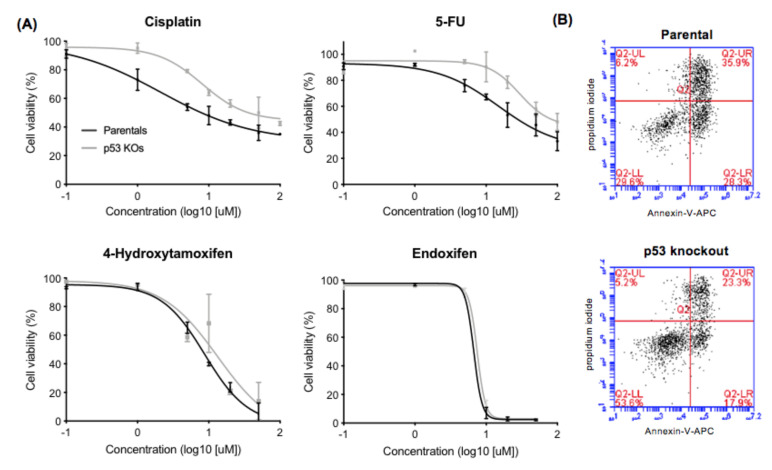
Cisplatin, 5-FU, 4-hydroxytamoxifen and endoxifen dose response curves. (**A**): Dose response curves for pooled parental and p53-KO cells treated with cisplatin, 5-FU, 4-hydroxytamoxifen, and endoxifen at doses between 1 µM and 100 µM (cisplatin and 5-FU) and 0.1 µM and 50 µM (4-hydroxytamoxifen and endoxifen) for 72 h. (**B**): Example of the Annexin-V-APC apoptosis assay following 50 µM cisplatin treatment (72 h) used to generate the dose response curves in (**A**). The population of p53-KO viable cells (lower left quadrant) following cisplatin treatment was higher than the parental population (53.6% vs. 29.6%) while the (early and late) apoptotic population was higher for the parental cells (28.3% (early apoptosis) and 35.9% (late apoptosis) vs. 17.9% and 23.3%). See Appendix A for examples of Annexin-V-APC apoptosis assays following 5-FU, 4-hydroxytamoxifen and endoxifen treatment.

**Figure 3 ijms-22-05547-f003:**
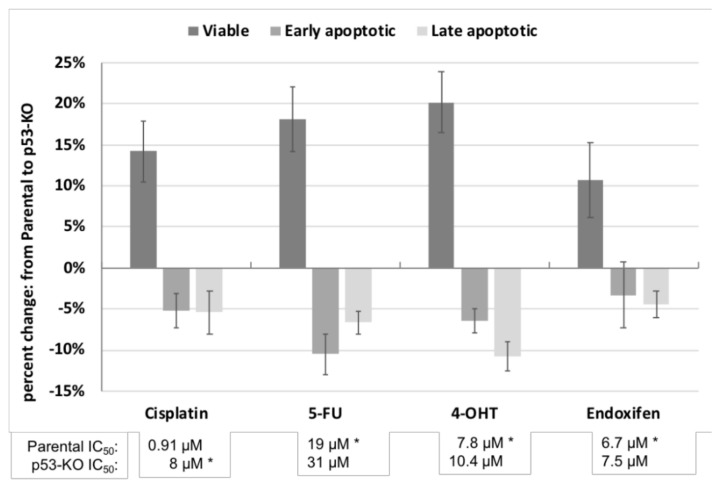
Response to cisplatin, 5-FU, 4-hydroxytamoxifen, and endoxifen. Percent change in cell viability and proportions of early and late apoptotic cells in p53-KO cells relative to parental cells, after drug treatments. IC_50_ concentrations for each drug are listed underneath the x-axis for parental and p53-KO cells. The drug concentrations used for the treatments are indicated with a star (*). The experiments were repeated three times. Error bars are standard errors. 4-OHT = 4-hydroxytamoxifen.

**Figure 4 ijms-22-05547-f004:**
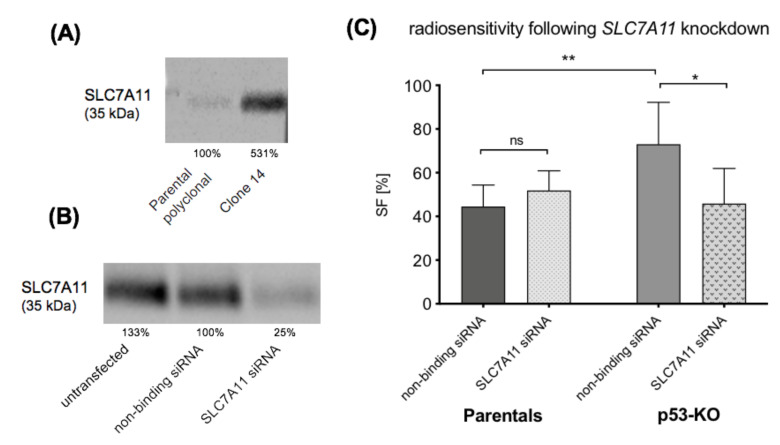
*SLC7A11* knockdown restores radiosensitivity in p53-KO cells. (**A**): SLC7A11 expression is higher in p53-KO than in parental cells. The % value represents the normalized expression (for protein quantification, protein expression was normalized to total protein load, see 4.7. for further details). (**B**): Western blot results confirm *SLC7A11* knockdown. *SLC7A11* knockdown of pooled p53-KOs cells (Clone 1, 7 and 14) resulted in 75%- reduced SLC7A11 expression. (**C**): The graph shows summarized results from the clonogenic assay from two independent irradiation experiments following *SLC7A11* knockdown. The latter sensitized the p53-KO cells to 2 Gy irradiation. SF: 46% vs. 73%, *p* = 0.0239. ns = non-significant, * 0.01 ≤ *p* < 0.05, ** 0.001 ≤ *p* < 0.01.

**Table 1 ijms-22-05547-t001:** miRNAs that were differentially expressed (decreased or increased) between parental vs. p53-KO cells. miRNAs that were reported to (directly) regulate p53 are indicated with *; miRNAs that were reported to be (directly) regulated by p53 are indicated with ^.

Increased or Decreased in p53-KO	MiRNAs	Differential Expression in p53-KO (Fold Change)	*p*-Value (Limma; FDR = 10%)	Reference for Interaction with p53
increased	hsa-miR-27a-3p *	1.37	0.0003	[31]
	hsa-miR-24-3p *	1.24	0.0043	[32]
	hsa-miR-185-5p *	1.25	0.0070	
	hsa-miR-130b-3p ^	1.31	0.0103	[33]
	hsa-miR-181a-5p *	1.16	0.0304	[34]
decreased	hsa-miR-324-3p	0.73	0.0001	
	hsa-miR-345-5p	0.71	0.0057	
	hsa-miR-328-3p	0.62	0.0068	
	hsa-miR-146b-5p	0.64	0.0084	
	hsa-miR-210-3p ^	0.55	0.0101	[35]
	hsa-miR-140-3p	0.48	0.0117	
	hsa-miR-26a-5p ^	0.86	0.0144	[36]
	hsa-miR-324-5p *	0.73	0.0225	[37]
	hsa-miR-26b-5p	0.86	0.0312	
	hsa-miR-320a-3p ^	0.79	0.0461	[38]

**Table 2 ijms-22-05547-t002:** Biological Pathway. Selection of pathways in which the validated targets of miR-27a-3p, or of all increased miRNAs, were significantly enriched (=pathway *p*-value (corrected) *p* ≤ 0.05). Pathways that were only significant in miR-27a-3p target gene analysis are indicated with *****; pathways that were only significant when target genes for all miRNAs were analyzed are not indicated.

Pathway Name	Pathway Uploaded Gene Count	Genes in InnateDB for This Entity	Pathway *p*-Value (Corrected)
Cellular responses to stress	81	240	1.93 × 10^−12^
Pathways in cancer	94	329	8.30 × 10^−10^
p53 pathway *	25	47	4.20 × 10^−8^
p53 signalling pathway	31	68	4.45 × 10^−8^
EGFR1 *	114	472	1.46 × 10^−7^
Gene expression *	223	1118	4.13 × 10^−7^
Cell cycle *	42	126	1.30 × 10^−6^
Cyclins and cell cycle regulation	14	23	8.72 × 10^−6^
Direct p53 effectors *	40	129	1.61 × 10^−5^
Rb tumour suppressor/checkpoint signaling in response to DNA damage *	10	13	1.75 × 10^−5^
WNT signaling pathway *	42	140	2.08 × 10^−5^
Cellular response to hypoxia	13	25	1.72 × 10^−4^
Fatty acid, triacylglycerol and ketone body metabolism*	46	176	2.61 × 10^−4^
mTOR signaling pathway	22	62	3.03 × 10^−4^
Regulation of lipid metabolism by PPARalpha *	32	109	3.92 × 10^−4^
VEGF signaling pathway	46	183	6.06 × 10^−4^
TGF-beta signaling pathway	25	80	8.39 × 10^−4^
Signal Transduction	4	4	0.004
Nuclear receptor transcription pathway *	14	39	0.004
Intrinsic pathway for apoptosis	13	38	0.008
PTEN dependent cell cycle arrest and apoptosis	7	15	0.013
Apoptosis	35	155	0.013
Plasma membrane oestrogen receptor signaling	9	24	0.019
Signaling to RAS	10	31	0.032
Apoptotic signaling in response to DNA damage	6	14	0.034
Metabolism of lipids and lipoproteins *	97	554	0.040
p53-Dependent G1 DNA damage response	15	57	0.040
p53-Dependent G1/S DNA damage checkpoint	15	57	0.040
Pyruvate metabolism *	8	24	0.050

**Table 3 ijms-22-05547-t003:** Biological Pathway. Selection of pathways in which the validated targets of all decreased miRNAs were significantly enriched.

Pathway Name	Pathway Uploaded Gene Count	Genes in InnateDB for This Entity	Pathway *p*-Value (Corrected)
Gene expression	330	1118	2.85 × 10^−9^
Translation	68	145	3.11 × 10^−9^
Cell cycle	172	523	3.80 × 10^−8^
SRP-dependent cotranslational protein targeting to membrane	51	107	1.57 × 10^−7^
Cellular responses to stress	91	240	2.18 × 10^−7^
Apoptosis	65	155	4.26 × 10^−7^
p53 signaling pathway	34	68	1.04 × 10^−5^
DNA Replication	44	105	5.86 × 10^−5^
p53-Dependent G1 DNA damage response	28	57	1.08 × 10^−4^
p53-Dependent G1/S DNA damage checkpoint	28	57	1.08 × 10^−4^
Direct p53 effectors	50	129	1.44 × 10^−4^
Metabolism of proteins	191	678	1.77 × 10^−4^
Stabilization of p53	24	52	0.001
Metabolism	382	1535	0.002
Ribosome	48	137	0.002
Cell cycle checkpoints	42	117	0.003
Regulation of apoptosis	25	60	0.004
Intrinsic pathway for apoptosis	18	38	0.004
WNT signaling pathway	33	91	0.008
p53 pathway	20	47	0.009
Alpha-linolenic (omega3) and linoleic (omega6) acid metabolism	7	10	0.013
Alpha-linolenic acid metabolism	7	10	0.013
Caspase cascade in apoptosis	21	53	0.016
Linoleic acid metabolism	5	6	0.018
Fatty acid, triacylglycerol and ketone body metabolism	53	176	0.024
Apoptotic signaling in response to DNA damage	8	14	0.026
Regulation of lipid metabolism by PPARalpha	35	109	0.032
Internal ribosome entry pathway	9	18	0.038
Validated nuclear oestrogen receptor alpha network	22	64	0.051

**Table 4 ijms-22-05547-t004:** miRNAs that were significantly differentially expressed after *SLC7A11* knockdown in p53-KO cells.

Increased or Decreased after *SLC7A11* Knockdown	MiRNAs	Differential Expression after *SLC7A11* Knockdown (Fold Change)	*p*-Value (Empirical Bayes; FDR = 10%)
decreased	hsa-miR-331-3p	0.93	0.019
increased	hsa-miR-30a-5p	1.27	0.025
	hsa-miR-1274A	1.37	0.041
	hsa-miR-1274B	1.37	0.047

**Table 5 ijms-22-05547-t005:** Biological Pathway. Selection of pathways in which the validated targets of miR-30a-5p were significantly enriched.

Pathway Name	Pathway Uploaded Gene Count	Genes in InnateDB for This Entity	Pathway *p*-Value (Corrected)
Direct p53 effectors	19	129	0.003
Intrinsic pathway for apoptosis	9	38	0.008
Cellular responses to stress	25	240	0.011
ATM signalling pathway	6	18	0.012
Apoptosis	13	88	0.016
Apoptotic signalling in response to DNA damage	5	14	0.020
Plasma membrane oestrogen receptor signalling	6	24	0.028
p53 signalling pathway	10	68	0.034
Hypoxia and p53 in the cardiovascular system	5	20	0.041
Cell cycle	14	126	0.043
Gene expression	70	1118	0.046

## Data Availability

Datasets were deposited in the Gene Expression Omnibus (www.ncbi.nlm.nih.gov/geo) GSE171965.

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
