# Peer review of "Mutant p53 Mediates Sensitivity to Cancer Treatment Agents in Oesophageal Adenocarcinoma Associated with MicroRNA and SLC7A11 Expression"

_ijms, 2021, doi:10.3390/ijms22115547_

Round 1
Reviewer 1 Report
Comments:
The authors of the manuscript "Mutant p53 mediates sensitivity to cancer treatment agents in oesophageal adenocarcinoma associated with microRNA and SLC7A11 expression" generated a mut-p53 knockout oesophageal adenocarcinoma cell line model using CRISPR/Cas and investigated the impact of mut-p53 knockout on miRNA expression and resistance to radiochemotherapy.
All Western blots lack the protein size and, most importantly, expression of a housekeeping gene.
The description of the mut-p53 knockout oesophageal adenocarcinoma cell line model generated using CRISPR/Cas is completely missing in the materials and methods section. No reference exists, and the cited references do not mention the model. In addition, the model has not been characterized (sequencing of successful genome editing, mRNA-expression of TP53) except for expression of p53 on protein level. Does the knockout of expression of mut-p53 restores wildtype p53 expression (especially after irradiation)?
The introduction lacks information about SLC7A11 - Why did the authors choose to study this protein? What is the function of SLC7A11? In addition, the particular function of p53 after irradiation should be mentioned.
The investigation of clonogenicity alone is not sufficient to determine the susceptibility of a cell line to irradiation. The authors do not provide a reason for choosing colony formation assays as a readout after irradiation. While cell self-renewal determined by colony formation assays might be affected, the viability/proliferation might not. The authors should provide an additional (proliferation/viability) assay. In addition, the non-irradiated controls must be plotted on the graphs for comparison purposes (Figure 1 and 4).
Its not clear why the authors also used hydroxytamoxifen and endoxifen for single treatment. They argue that both drugs “augments the cytotoxic effects of cisplatin and 5-FU”.
Figure 2B belongs to Figure 3 or should be shown in the supplements. Annexin staining following all treatments should be shown. It would greatly improve the message of Figure 3 if the "true" values were shown rather than the % difference. Significances are missing. Control treatments for all cell lines should be shown.
The fold change of the “differentially” expressed miRNAs is very low. The authors in addition do not provide a single validation experiment. Does the increase of only 16% of miR-181a-5p or decrease of 14% of miR-26b-5p indeed might have an effect??? The chosen miRNAs miR-324-3p (at least is decreased by 27%) and miR-27a-3p (increased by 27%) are indeed highly significant. However, all subsequent studies are based on expression analysis, which has not been validated. The authors hypothesize that the selected miRNAs play a role in a variety of processes in their oesophageal adenocarcinoma cell lines. However, they validated neither the differential expression of the miRNA itself, nor the impact of this on the highly diverse processes. Does inhibition of e.g. miR-27a-3p in p53-KO cells or inhibition of miR-324-3p in the control cells indeed have an impact on the stated mechanism(s)?
As mentioned at the beginning, it is not clear why SLC7A11 was chosen. The exact function of SLC7A11 in dependence on p53 should be explained in more detail. The regulation of SLC7A11 by the three siRNAs (miR-26b-5p, -27a-3p -181a-5p) in p53-KO cells has not been validated and is only assumed but should be proofed.
Allover, the measured effects had been strongly interpreted without being validated. Almost the entire discussion relies on hypotheses based on weak differential expression of a few, non-validated miRNAs. Both the expression and the impact of the identified miRNAs needs to be validated in at least one example. Thus, the discussion can be greatly shortened as the authors can focus on their hypotheses after validation.
Author Response
Please see the attached word document

Reviewer 2 Report
Eichelmann et al describe the role of mutant p53 in a cell line model. KO of mutant p53 resulted in creased chemo and radioresistance and altered miRNA expression levels. Some of this miRNAs were direct regulators of SLC7A11, and they descrived how SLC7A11 ko restored radiosensitivity of p53-ko cells.
Results are interesting, but none of the western blots have loading control. It is not acceptable to determine the protein levels without using a loading control. It is very difficult to see if they really get a ko or compare reduced levels. Authors need correct this mistake in figures 1 and 4. Also, if possible, they should repeat the key experiments with another cell line to confirm the finding and avoid an effect of selected clones from a specific cell line. It will definitely strength the results obtained.
Author Response
Please see the attached word document

Round 2
Reviewer 1 Report
Dear authors,
Thank you very much for your detailed answers and explanations, which improved the quality of the manuscript.
However, I do not agree with all of your points:
First, the sensitivity of the cells to irradiation should be determined by cell proliferation and clonogenicity assays. Of note, both assays might give different results since proliferation and self renewal ability are differential regulated processes that might be affected by irradiation. Thus, the addition of a proliferation test might be more convincing.
Second, and most importantly, although I am not an expert on miRNAs, it seems to me that it is common to validate the biological function of miRNAs using various techniques (see, for example, abstract DOI: 10.1038/ng1799). It would increase the quality of the manuscript if some miRNAs were overexpressed in parental cells, for example, and sensitivity to irradiation, cell cycle effects, or other effects that the authors currently only suspect were examined. Without validation and given that the observed fold changes are quite small, the conclusions drawn might seem somewhat over-interpreted.
Reviewer 2 Report
Authors have clarified the points raised during the revision process and the manuscript has been improved.
Round 3
Reviewer 1 Report
Dear authors, thank you very much for clarification.